# Intra-Articular Hyaluronic Acid in Osteoarthritis and Tendinopathies: Molecular and Clinical Approaches

**DOI:** 10.3390/biomedicines11041061

**Published:** 2023-03-30

**Authors:** Fabio Ramos Costa, Mariana Ramos Costa Marques, Vinicius Calumby Costa, Gabriel Silva Santos, Rubens Andrade Martins, Marcia da Silva Santos, Maria Helena Andrade Santana, Arulkumar Nallakumarasamy, Madhan Jeyaraman, João Vitor Bizinotto Lana, José Fábio Santos Duarte Lana

**Affiliations:** 1Department of Orthopaedics, FC Sports Traumatology Clinic, Salvador 40296-210, Brazil; 2Department of Orthopaedics, Brazilian Institute of Regenerative Medicine, Indaiatuba 13334-170, Brazil; 3Medical School, Tiradentes University Center, Maceió 57038-800, Brazil; 4Department of Nutritional Sciences, Metropolitan Union of Education and Culture, Salvador 42700-000, Brazil; 5School of Chemical Engineering, The University of Campinas, Campinas 13083-852, Brazil; 6Department of Orthopaedics, All India Institute of Medical Sciences, Bhubaneswar 751019, Odisha, India; 7Department of Orthopaedics, ACS Medical College and Hospital, Dr MGR Educational and Research Institute, Chennai 600056, Tamil Nadu, India; 8Department of Biotechnology, School of Engineering and Technology, Sharda University, Greater Noida 201310, Uttar Pradesh, India; 9Medical School, Max Planck University Center, Indaiatuba 13343-060, Brazil

**Keywords:** hyaluronic acid, orthopedics, orthobiologics, inflammation, viscosupplementation, regenerative medicine

## Abstract

Musculoskeletal diseases continue to rise on a global scale, causing significant socioeconomic impact and decreased quality of life. The most common disorders affecting musculoskeletal structures are osteoarthritis and tendinopathies, complicated orthopedic conditions responsible for major pain and debilitation. Intra-articular hyaluronic acid (HA) has been a safe, effective, and minimally invasive therapeutic tool for treating these diseases. Several studies from bedside to clinical practice reveal the multiple benefits of HA such as lubrication, anti-inflammation, and stimulation of cellular activity associated with proliferation, differentiation, migration, and secretion of additional molecules. Collectively, these effects have demonstrated positive outcomes that assist in the regeneration of chondral and tendinous tissues which are otherwise destroyed by the predominant catabolic and inflammatory conditions seen in tissue injury. The literature describes the physicochemical, mechanical, and biological properties of HA, their commercial product types, and clinical applications individually, while their interfaces are seldom reported. Our review addresses the frontiers of basic sciences, products, and clinical approaches. It provides physicians with a better understanding of the boundaries between the processes that lead to diseases, the molecular mechanisms that contribute to tissue repair, and the benefits of the HA types for a conscientious choice. In addition, it points out the current needs for the treatments.

## 1. Introduction

Hyaluronic acid (HA), commonly referred to as hyaluronan, is a natural biological compound present in many tissues and fluids [1]. HA was first isolated as glycosaminoglycan (GAG) in 1934 by Meyer and Palmer from bovine vitreous humor. The term “hyaluronic acid” is broken down into hyaloid, which means vitreous, and uronic acid [1]. Posteriorly, HA was identified in other organs and tissue types, such as skin, joints, and the human umbilical cord, to name a few. Researchers discovered that this product could also be synthesized by many bacterial species such as *Escherichia coli, Bacillus subtilis*, and *Streptococcus zooepidemicus* via fermentation [2]. Conveniently, the chemical structure and properties of HA are identical in both vertebrates and bacteria [1]. More importantly, it was found to be produced by various cell types in the body during different stages of the cell cycle, being a major component of the extracellular matrix (ECM) [3]. In humans, HA has been considered for the treatment of painful musculoskeletal conditions such as tendinopathies and degenerative disorders like osteoarthritis (OA) [4,5]. HA has been utilized for the management of several health conditions since the 1970s due to its unique physicochemical properties and biological functions, as this molecule establishes strong interactions with cells and ECM [3,6]. The objective of this review is to discuss the biological functions of hyaluronic acid and its regenerative medicine potential in the treatment of osteoarthritis and tendinopathies.

## 2. Etiopathogenesis of Osteoarthritis

Osteoarthritis (OA), one of the most common degenerative and progressive joint diseases, is a major cause of pain and disability in adult individuals, affecting approximately 7% of the global population [7]. The numbers have been increasing significantly in the past few decades [8], and this is likely attributed to factors such as the aging of the population and the incidence of poor health status, especially metabolic syndrome [9,10,11]. OA is influenced by the complex interplay between local, systemic, and external factors, which consequently dictate the progression outcome and the patient’s response to treatment [12]. The most notable features include progressive loss of articular cartilage, osteophyte formation, subchondral bone thickening, escalated synovial inflammation, ligament and meniscal deterioration, and overall joint hypertrophy [10].

OA has been observed to arise from a confluence of diverse elements, encompassing genetic predisposition, adiposity, trauma, senescence, and even the co-occurrence of additional systemic maladies [13]. This condition harms the complete joint complex, and earlier studies have indicated that the degenerative process takes place in two discernible stages. Firstly, in the anabolic phase, chondrocytes make multiple efforts to mend injured ECM. Subsequently, in the catabolic phase, increased activity of enzymes that break down molecules leads to ECM digestion and hindrance of fresh ECM creation [14]. Prolonged biomechanical and biochemical strain induces secondary modifications, culminating in a preponderant shift towards catabolic reactions. These physiological mechanisms are accountable for the wearing away of cartilage and damage to the subchondral bone and surrounding structures, exacerbating physical discomfort and disability [15].

To elaborate, synoviocytes and osteoarthritic chondrocytes generate high amounts of MMPs (matrix metalloproteinase) 1, 3, 9, and 13, at least in vitro [16]. Synoviocytes secrete proteolytic enzymes and pro-inflammatory cytokines such as IL-(interleukin) 1β, IL-6, and tumor necrosis factor-alpha (TNF-α), which additionally impact OA progression and the perception of noxious stimuli associated with the disease [17]. Other molecules, including resistin and osteopontin, are linked to the severity of the disease and exhibit a notable increase in expression within the osteoarthritic synovial tissue [18,19,20]. In addition, it has been reported that the synovium alone is capable of producing certain chemokines and metalloproteinases that contribute to the degeneration of cartilage, even though cartilage itself also generates most of the catabolic molecules through autocrine and paracrine signaling mechanisms [21]. As a result, the remnants generated from cartilage degradation, whether through mechanical or enzymatic disintegration, may elicit the discharge of collagenase and other hydrolytic enzymes from synovial cells. This sequence of events instigates vascular hyperplasia in synovial membranes affected by osteoarthritis [22].

Within the joints of healthy adult individuals, the articular cartilage is mainly composed of chondrocytes and ECM, which comprises various substances, including water, chondroitin sulfate, type II collagen, proteoglycans, HA, as well as other proteins such as fibronectin and laminin. Additionally, the fibrous components of the ECM contain elastin and collagen, which consist of several types of fibrillar collagens such as types I, II, III, V, and XI, as well as non-fibrillar collagens including FACIT types IX, XII, and XIV, short chain types VIII and X, and basement membrane type IV [23,24]. The rate of collagen turnover is rather slow whereas that of the proteoglycan is relatively faster [15]. This process is controlled by chondrocytes, which are responsible for the synthesis of molecular components including proteolytic enzymes that regulate their breakdown [15]. These cells are also exposed to multiple sources of stimuli, including polypeptides, cytokines, biomechanical signals, and even fragmented components of the ECM itself [15].

Osteoarthritis occurs due to dystrophic damage to the articular cartilage in response to an imbalance between anabolic and catabolic reactions in the chondral and subchondral bone compartments [15,25]. Factors such as metabolic syndrome, physical trauma, microfractures, and inflammation contribute to a slight increase in enzymatic activity, resulting in the formation of “wear” particles, hence, the so-called “wear-and-tear” process [11,26]. Molecularly, the dysregulated proteoglycan metabolism destabilizes collagen fibers, leading to dehydration and disorganization of cartilage. Increased degradation of glycosaminoglycans, especially chondroitin sulphate and hyaluronic acid, leads to a decrease in matrix resistance to biomechanical stress [15]. This also increases the sensitivity of the cartilage surface to damage. Excessive synthesis and local release of MMPs by chondrocytes gradually delay cartilage repair. Collectively, these reactions contribute to softening, fibrillation, ulceration, and ultimately, the destruction of articular cartilage [15].

At the cellular level, macrophages may very well phagocytize microparticles and cellular debris, eventually. However, the overproduction of these particles causes significant cell stress, making it harder to dispose of them. Eventually, they assume the role of mediators of inflammation, eliciting chondrocytes to secrete elevated quantities of catabolic enzymes [26]. Molecules arising from the breakdown of collagen and proteoglycan are additionally subjected to processing by synovial macrophages, prompting the release of TNFα, IL-1, and IL-6. Subsequently, these molecules bind to chondrocyte receptors, stimulating further MMP discharge and the hindrance of collagen type 2 synthesis. This sequence of events aggravates cartilage degeneration, favoring a more debilitated microenvironment [27]. In summary, disturbance of homeostasis leads to an elevation in water content and a reduction in proteoglycan content within the ECM. This undermines the integrity of the collagen framework due to decreased synthesis of collagen type 2 and the amplified degradation of pre-existing collagen. Ultimately, this culminates in an increased rate of chondrocyte apoptosis [28].

## 3. Etiopathogenesis of Tendinopathy

Tendinopathies are a common type of pathology amongst the general population and are even more frequent in sportsmen. Tendon disorders range from traumatic injuries to chronic disease processes [29]. They are one of the most frequent orthopedic diagnoses and account for approximately 30% of musculoskeletal cases [30]. Every year, at least 30 million medical procedures involving tendons take place worldwide, generating a major socioeconomic impact on affected individuals [31]. It is estimated that at least 50% of sports-related injuries involving tendons are attributed to overuse conditions causing significant physical stress on these tissues [32]. The most frequently affected anatomical sites include the long head of the brachial biceps, extensors, and flexors of the wrist, rotator cuff, patellar tendon, tibial tendon, thigh adductors, and the Achilles tendon [29,32].

In addition to sports-related injuries, other intrinsic and extrinsic risk factors such as aging, biological sex, and even poor workplace ergonomics can contribute to the development of specific tendinopathies [33,34]. Prolonged use of certain pharmacological substances can be quite detrimental to tendon tissue biology and render individuals more susceptible to this disorder. Intrinsic mechanisms can also play a role in the progression of tendon pathology, especially in terms of metabolic dysregulation [35]. In patients with hypercholesterolemia, for instance, cholesterol deposition in tendons can mark the establishment of chronic low-grade inflammation and degeneration of tendons over time. Similarly, the glycation end-products in diabetic patients also deteriorate the biological properties and mechanical function of not only tendons but all musculoskeletal tissues in general [11,36,37]. Another notorious hallmark of metabolic syndrome is the accumulation of visceral fat and body weight. Elevated mechanical effort applies more biomechanical stress to weight-bearing joints, leading to the deterioration of musculoskeletal structures [38].

On the molecular level, tendons are organized according to a hierarchical structure. Collagen protein subunits make up the smallest building blocks of the tendon, which combine to form tropocollagen helixes [39]. Tendons contain types I, II, and III collagen, elastin, water, and proteoglycans, such as decorin [40]. Type I collagen forms stiff structures that convey mechanical durability and strength, whereas type II collagen has comparably smaller fibrils. Generally associated with scar tissue and injury, type III collagen gives rise to thinner fibers which are mostly concentrated in the skin, blood vessels, and other tissues which contain high amounts of elastic fibers [40].

These structures also have a cellular component and an ECM as inherent parts of their structure. Tenoblasts are immature spindle-shaped cells that ultimately differentiate into tenocytes upon reaching maturation. These two cells are the most abundant cell types present in tendons and play a key role in producing ECM, which must contain proteoglycan, collagen, and fibronectin, the essential proteins for tendon homeostasis and regeneration [41]. A small percentage of different cell types also populate tendinous structures, including capillary endothelial cells, arteriolar smooth muscle cells, synovial cells, and chondrocytes [42]. All of these cells are surrounded and supported by the ECM, which is a complex organic structure containing mostly collagen, elastin, fibrillin, fibronectin, and proteoglycans [43].

Both the cellular and molecular components in tendon health are in constant interplay. The production of various biomechanical and biochemical signals elicits a wide set of responses that may be either beneficial or detrimental depending on the patient’s overall health status [11,44]. The activation of certain cell signaling pathways may or may not change the ECM function and composition. The Scx and Mkx pathways, for example, interact with Smad3. This molecule is a key transcriptional mediator of TGF-β signaling to control ECM synthesis in tendinous tissues [45]. In addition to regulating ECM production, the Mkx pathway also controls tendon maturation. Therefore it may partially contribute to the maintenance of tenocytes by impeding their ability to undergo differentiation into other cell fates such as myogenic or skeletogenic phenotypes [45]. It is also worth noting that signaling pathways activated by mechanical loading can increase the expression of Scx, Mkx, and Smad3, stimulating the production of more ECM [45].

The maintenance of tendon homeostasis is primarily governed by mechanical loading, which is subsequently modulated by cellular activity under the influence of neuronal and cellular mediators. Mediators may be released locally or remotely and then carried away through either blood circulation or nerve supply [46,47]. Considering these facts, impaired biomechanics and dysregulated cellular processes may be recognized as the major culprits in tendon pathology. Proper loading of tendons is what stimulates anabolic responses, especially the upregulation of collagen type I gene expression and increased synthesis of these proteins [48,49]. Type I collagen synthesis peaks approximately 24 h following physical modulation and its levels are kept high for up to 80 h. Nevertheless, an excess of physical impact can trigger the deterioration of collagen proteins and consequently provoke a prevailing catabolic response. However, the occurrence of the catabolic peak precedes the anabolic response in terms of timing. This results in a net reduction of collagen within the initial 24 to 36 h after exercise, followed by a subsequent net gain [50]. This implies that sufficient rest periods between physical activity sessions are crucial for maintaining a healthy tendon homeostasis. Should these limitations not be met due to excessive or repetitive loading, tenocytes are then forced to produce inflammatory molecules. This in turn fragilizes collagen fibrils and increases the risk of microdamage [51].

Tendons subjected to repetitive mechanical loading have been found to exhibit an exacerbated production of inflammatory cytokines, including prostaglandin E2 (PGE2) [52]. An animal study shows that injections of PGE2 in leporine cause degenerative alterations in the tendon proper [53], whilst peritendinous injections of PGE1 lead to tendinopathy in murines [54]. The inflammatory response in tendons is prominently characterized by pro-inflammatory biomarkers, including but not limited to, IL-18, IL-15, IL-6, IL-1β, and TNF-α [55,56], which are associated with the activation of the prototypical pro-inflammatory signaling pathway, nuclear factor kappa B (NF-κB) [57]. Moreover, in models of Achilles tendinopathy, it is also common to observe granulation changes in capillary vessels as well as the infiltration of inflammatory cells such as macrophages, mast cells, and B and T lymphocytes [58], illustrating the regulatory role of the innate immune system in the early onset of the disease. It is worth noting that the role of macrophages during inflammation and tissue repair is vital. Signaling pathways can induce the polarization of macrophages into either M1 subtype (pro-inflammatory) or M2 subtype (anti-inflammatory) [59,60]. The differentiation of monocytes and polarization of macrophages are regulated by various inflammatory mediators, including interferons, NF-κB, and glucocorticoid receptor activation pathways [60]. Therefore, inflammatory pathways in tendon disorders control macrophage polarization, leading to failed, fibrotic healing responses [61].

Repetitive mechanical stress can cause tenocytes and fibroblasts to bind transforming growth factor β (TGF-β) and pro-inflammatory cytokines, leading to their differentiation into myofibroblasts [62]. Once the healing process is completed, the mechanical stress on myofibroblasts is removed, and these cells undergo apoptosis. The problem begins when this mechanism fails, as myofibroblasts then trigger a hyperproliferative process that culminates in fibrosis, a major histological feature of tendinopathy [63].

Angiogenesis is a critical event in the healing process and is regulated mainly by the vascular endothelial growth factor (VEGF), which stimulates endothelial cell migration via chemotaxis and vasodilation. However, in tendinopathy, neovascularization can lead to the deterioration of mechanical properties and even ruptures. Furthermore, the sprouting and ingrowth of sensory nerve fibers following the neoangiogenic process into the tendon can trigger nociception and pain in patients with tendinopathy [58]. The uncontrolled and aberrant sprouting of sensory nerve fibers into the tendon during tendinopathy is indicative of a failed healing response, which can lead to increased pain signaling. This process also plays a role in the hyperproliferative changes observed in tendinosis [63]. Many interventional therapies have been proposed for both OA and tendinopathies. Conservative methods such as the administration of pharmacological agents only bring temporary alleviation of pain but do not address the root of the pathology [64,65]. Physicians may prescribe a combination of drugs at different stages of OA, aiming to block inflammatory nociceptive pain. Non-steroidal anti-inflammatory drugs (NSAIDs), other analgesics, and corticosteroids, for example, are commonly indicated for the management of pain. However, prolonged administration of NSAIDs is known to cause secondary health problems. Although they may effectively target pain, their prolonged use can lead to serious side effects such as peptic ulcer disease, acute renal failure, and myocardial infarction [66]. Non-pharmacological alternatives such as physical therapy, low-impact exercise, weight loss, physical aids, and nerve ablation are often recommended to minimize the risks associated with NSAIDs. Even so, in severe cases such as grade 4 OA or complete tendon rupture, invasive procedures may be inevitable, causing a significant impact on the quality of life [12,64].

## 4. Physicochemical and Biological Properties of Hyaluronic Acid

The physicochemical properties of HA are given by molecular mass and spatial conformation, classifying it as high (HMW), medium (MMW), or low molecular weight (LMW) [67]. The molecular weight of HA seems to be a crucial factor regarding the biological functions it elicits in human tissues. Typically, LMW HA falls in the range of 500,000 to 730,000 Daltons (Da), MMW in between 800,000 and 2,000,000, and HMW with an average of 6,000,000 [68]. In the joints of healthy individuals, HA has a mass of about 5 to 7 million Da, whereas in osteoarthritic joints, its mass falls to approximately 1 million Da. The crosslinking of HMW HA molecules forms a solution with high viscosity, serving as a shock absorber and lubricant. Additionally, this type of HA has properties that support the growth of cells [69]. HA is the primary non-protein component of synovial fluid, and it surrounds cells as a layer. Whether natural or artificial, it interacts with pro-inflammatory mediators and binds to cellular receptors, regulating cell proliferation, migration, and gene expression [70]. HA is also a potent collagen stimulator (especially type I collagen), and is thus capable of promoting tissue recovery and maintenance of cellular integrity [71]. These properties make HA a particularly useful orthobiologic tool for the healing of tendon and chondral tissues in many circumstances.

The IA injection of HA is considered a minimally invasive intervention method that has not been associated with any significant systemic adverse events, unlike other types of IA injections, such as corticosteroids, or even oral administration of NSAIDs [72]. Studies have shown that this alternative approach has demonstrated positive effects in vitro. The ECM has been found to have an impact on cell metabolism, particularly on osteoblasts in subchondral bone affected by osteoarthritis, and HA has been shown to reverse abnormal homeostatic activity [73]. As Figure 1 suggests, IA-HA has been demonstrated to have the potential to reduce chondrocyte apoptosis, as well as increase chondrocyte proliferation [74]. In humans, it is important to use formulations with medium to high MWs to mimic the conditions and biological properties of HA naturally produced in the body. In addition, it is important to use HA derived from biological synthesis, to avoid undesired side effects [67].

Administration of LMW HA results in weak binding and therefore weak HA biosynthesis. With MMW HA, there is stronger binding and a higher number of HA receptors being stimulated, which enhances endogenous HA production. On the other hand, it must be emphasized that extremely large molecules present in HMW HA products may not always be convenient as the large domains of these molecules can limit the number of free binding sites on the cell surface, which logically implies a relatively weak stimulation of HA biosynthesis [68]. Despite these physiological hurdles, practitioners must not forget that MW is not the only limiting factor because HA concentration in IA approaches is also another variable with its fair share of influence on clinical outcomes.

The balance in HA turnover has a key role in determining its concentrations, MW, and, consequently, the properties it will display in disease processes. Formulations with higher molecular weights usually elicit anti-inflammatory effects, as it regulates the recruitment of immune cells. Conversely, HA with lower molecular weights is known to promote angiogenesis and tissue remodeling in wound healing but it may also exhibit a more pro-inflammatory activity in specific cell types such as chondrocytes [75,76]. Endogenously, HA synthesis by synovial fibroblasts is influenced by MW and HA concentration in the extracellular environment [68].

The protective effects of this modality are attributed to the ability of HA to bind to CD44 receptors. This inhibits the expression of IL-1β, which dampens the production of MMPs 1, 2, 3, 9, and 13 [77,78,79], impeding catabolic enzyme activity within musculoskeletal tissues [80]. Upon interacting with surface receptors, HA activates intracellular signaling systems involved in the proliferation, differentiation, migration, and degradation of HA itself as shown in Figure 2 [81]. CD44 is the most widely studied HA receptor as it is expressed in nearly all human cell types. The affinity of CD44 for HA is what determines the potential of HA as a signaling molecule. However, this depends both on the concentration and molecular weight of HA, by glycosylation of extracellular domains and phosphorylation of serine [82]. CD44 is clustered by HMW HA polymers and can interact with other ligands, including growth factors, ECM proteins, MMPs, and cytokines [83]. Another important HA surface receptor is RHAMM, commonly referred to as CD168. It is expressed in many different tissues and regulates cell migration by interacting with skeletal proteins, especially in the healing cascade [82]. The interaction between HA and CD168 plays a major role in the activation of signaling pathways that involve Src and other kinase protein complexes of focal adhesions [84].

The application and benefits of HA for several orthopedic conditions have been well documented in the literature, from basic laboratory studies to robust clinical trials and systematic reviews. Viscosupplementation with HA and HA-derived biomaterials has been propelling major advances in the treatment of OA since it was initially proposed in the 1970s by Endre A. Balazs [82]. Posteriorly, in the 1980s, new HA derivatives emerged as suitable IA injection strategies envisaging the restoration of joint homeostasis and protection against mechanical damage [85].

The most widely used HA products in clinical trials were Synvisc and Hyalgan due to their safety, efficacy, and long-lasting effects despite the need for IA injections [82,86]. Hyalgan, in particular, has been shown to enhance the survival and proliferation of human chondrocytes which are exposed to reactive oxygen species (ROS) [87]. Since then, many additional HA derivatives with different MWs were made available as depicted in Table 1, which was created based on the research developed by Migliore et al. [88]. More recently, an alternative product consisting of the mixture of HA and lactose-modified chitosan (Chitlac^®^) has shown promising results in improving the anti-inflammatory effect and therapeutic value of HA in OA. In vitro and in vivo studies have shown an expressive increase in cartilage regeneration after administration of this derivative in experimentally-induced OA [89,90]. Its effects have been further revealed in a recent study published in 2021. The HA-Chitlac^®^ mixture significantly attenuates triamcinolone acetonide-hydroxypropyl-β-cyclodextrin (TA-CD) drug cytotoxicity in human chondrocyte cultures and sustains anti-inflammatory effects, reinforcing yet again the chondroprotective role of the HA for OA [91]. In more advanced and severe stages, HA alone may not suffice, requiring additional interventions such as autologous chondrocyte implantation. Three-dimensional biodegradable and biocompatible HA-based scaffold polymers such as Hyaff-11^®^ have been successfully utilized in the past for human chondrocyte cultures [92]. Once chondrocytes are implanted, the regenerated tissue undergoes a process of maturation and forms hyaline tissue instead of fibrous cartilage [92].

A randomized controlled trial [93] compared the clinical effects of platelet-rich plasma (PRP) and HA individually and synergistically for mild to moderate degrees of knee OA in 105 patients. The patients were randomly allocated to HA, PRP, or HA+PRP groups and received 3 intra-articular knee injections of their assigned substance with 2-week intervals between each infiltration. Clinical outcomes were evaluated according to Western Ontario and McMaster Universities Arthritis Index (WOMAC) and the Visual Analogue Scale (VAS) questionnaires at baseline and after 1, 3, 6, and 12 months. In this study, the association of HA and leukocyte-rich PRP was found to be statistically significant in terms of clinical outcome as physical function and pain were dramatically improved in the first 30 days after treatment. These observations may be attributed to the fact that HA provides a functional matrix with supportive scaffold properties enhancing cartilage biomechanics and tissue repair [94].

The exact mechanisms of action regarding exogenous HA administration are complex. However, current hypotheses suggest that HA can relay its known protective effects in 2 distinct stages. The first stage is known as the mechanical stage, where synovial fluid is substituted by higher concentrations of HA to improve viscosity [70]. Additionally, this allows the restoration and improvement of lubrication, and shock-absorbing properties of synovial fluid and establishes a layer around nociceptors that reduces pain signaling [95]. The second and final stage is often referred to as the pharmacological stage, where biosynthesis of endogenous HA and ECM components takes place [96]. This reduces the scarcity of proteoglycans in cartilage and prevents chondrocyte apoptosis [97]. Moreover, it also dampens inflammatory cell activity, reducing HA degradation and production of nociceptive mediators [70].

Similarly, the viscoelastic properties of HA have also promoted beneficial effects in numerous in vitro studies of tendinous tissue without significant adverse effects [98,99,100,101,102,103,104,105,106,107]. Most of the early in vitro studies [98,99,100,101] aimed at investigating the effects of various HA coating solutions on the gliding resistance of tendons. Collectively, the findings show that HA coating of tendons does not significantly improve gliding resistance compared to saline controls, although their surface becomes significantly smoother, in contrast. In the in vitro study led by Taguchi et al. [100], the researchers attempted to replicate these experiments in human tendons. Much like the animal models, similar observations were reported. Saline solutions promoted significantly higher gliding resistance in comparison to HA treatment, thus suggesting that HA may limit tendon adhesion to some extent.

In regard to tendon healing, however, the results were relatively more optimistic. In a few models of tendon injury [102,103,104], HA did not significantly increase the expression of procollagen alpha 1 but was still able to bind to its receptors and induce cell proliferation in dose-dependent manners, likely by HA sensitivity. Furthermore, one study on human tendons derived from rotator cuff tears found that HA increases cell viability and proliferation at 24 h compared to controls, regardless of molecular weight [108]. In contrast to the previous studies, the synthesis of collagen type I was stimulated at 14 days, being significantly higher in the HMW HA group. Speaking of which, another investigation [109] found that an HMW HA formulation with a weight greater than 1.2 megadaltons inhibits activation of the NF-kB inflammatory pathway caused by advanced glycation end products. Conversely, LMW HA does not convey benefits and is associated with pro-inflammatory status [75,76].

A parallel study led by Tanimoto et al. [106] evaluated the effects of pro-inflammatory cytokines TNF-α and IL-1β on rabbit HA-synthetase (HAS) mRNA expression. Under inflammatory conditions typically seen in wound healing, these cytokines increase HAS mRNA expression and may therefore contribute to the fragmentation and accumulation of HA. Similarly, Smith and colleagues [107] demonstrated that the exogenous addition of HA to synovial fibroblast cultures stimulates HA synthesis according to increasing concentrations and MW.

In regards to cost-effectiveness and comparison with other known treatments for orthopedic conditions, a recently published study has compared IA administration of HA versus PRP for the treatment of symptomatic knee osteoarthritis [110]. Samuelson and colleagues found that the cost per quality-adjusted life-year (QALY) of a series of PRP injections was USD 8635.23/QALY, whereas HA injections were USD 5331.75/QALY; however, PRP was significantly more effective at 1 year.

Rosen et al. [111] aimed to compare IA-HA with conservative treatments (physical therapy, orthosis, NSAIDs, and analgesics) for early to moderate-stage knee osteoarthritis. The authors revealed that HMW HA was superior to LMW HA and physical therapy, being less expensive yet providing greater benefits. HMW HA was cost-effective in comparison to orthosis and NSAIDs/analgesics.

## 5. Conclusions

Hyaluronic acid is an essential compound that can be naturally found in many organs and tissues. This molecule plays a vital role in musculoskeletal health, especially in painful conditions such as osteoarthritis and tendinopathies. Intra-articular administration of hyaluronic acid as an orthobiologic tool is a minimally invasive procedure with demonstrated efficacy and safety. This therapeutic alternative offers multiple benefits associated with attenuated inflammation, lubrication, improved biomechanics, cell proliferation, differentiation, migration, and enhanced protein biosynthesis and secretion. Despite the growing number of new HA derivatives for the treatment of orthopedic diseases, future investigations are still needed to further comprehend the factors that contribute to musculoskeletal tissue repair.

## Figures and Tables

**Figure 1 biomedicines-11-01061-f001:**
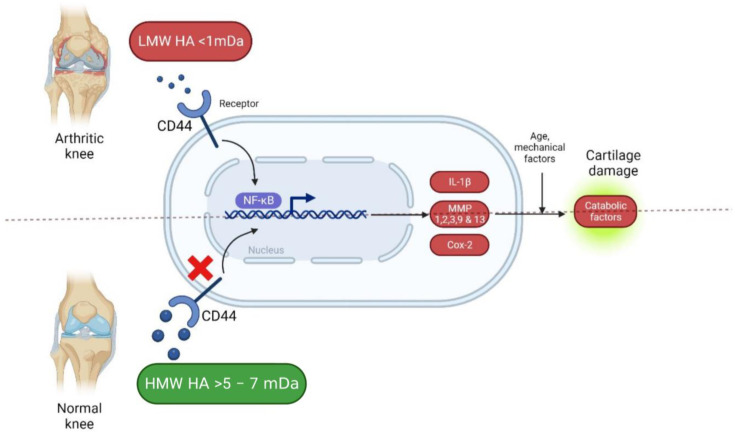
Hyaluronic acid in arthritis knees (Created with Biorender.com). [https://app.biorender.com/, accessed on 20 December 2022].

**Figure 2 biomedicines-11-01061-f002:**
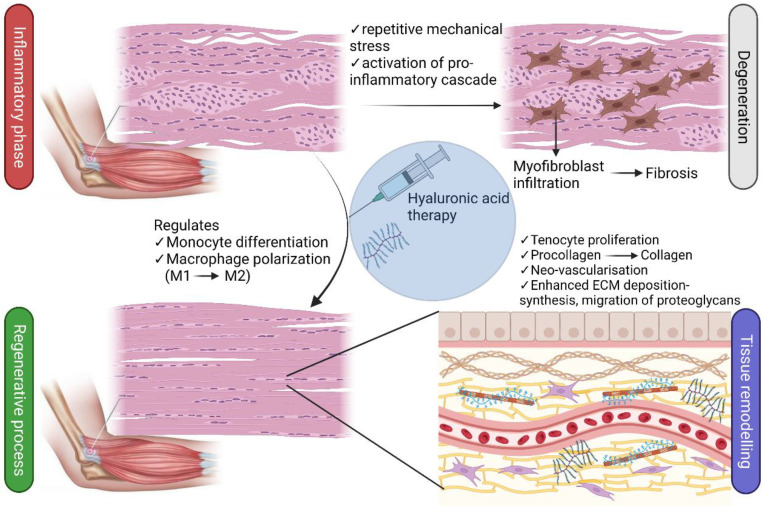
Hyaluronic acid in tendinopathies (Created with Biorender.com). Hyaluronic acid in orthopedic conditions. [https://app.biorender.com/, accessed on 20 December 2022].

**Table 1 biomedicines-11-01061-t001:** Typical hyaluronic acid formulations in orthopedics.

Brand	Source	Concentration	Molecular Weight	Indication
Synolis^®^ VA	Biofermentation	HA: 20 mg (2%) Sorbitol: 40 mg (4%)	High molecular weight (2.1 MDa)	Knee or Hip Osteoarthritis
RenehaVis^®^	Biofermentation	Low molecular weight: 15.4 mg (2.2%); High molecular weight 7 mg (1%)	Low molecular weight (<1 MDa) and High molecular weight (2 MDa)	Knee or Hip Osteoarthritis
SportVis^®^	Biofermentation	12 mg (1%)	Not reported	Soft tissues (tendon injuries)
Ostenil^®^	Biofermentation	20 mg (1%)	High molecular weight (1–2 MDa)	Osteoarthritis of shoulders, hips, and knees
Ostenil^®^ Plus	Biofermentation	HA: 40 mg (2%) Mannitol: 10 (0.5%)	High molecular weight (1–2 MDa)	Knee Osteoarthritis
OrthoVisc^®^	Biofermentation	30 mg (1.5%)	High molecular weight (1.1–2.9 MDa)	Knee Osteoarthritis
OrthoVisc^®^ mini	Biofermentation	15 mg (1.5%)	High molecular weight (1.4 MDa)	Small joints
MonoVisc^®^	Biofermentation	80 mg	High molecular weight (1–2.9 MDa)	Knee Osteoarthritis
Synvisc^®^	Rooster Comb	16 mg (80% HA HMW cross-linked; 20% gel cross-linked)	High molecular weight (6 MDa)	Osteoarthritis of shoulders, hips, and knees
Synvisc^®^ One	Rooster Comb	48 mg (80% HA HMW cross-linked; 20% gel cross-linked)	High molecular weight (6 MDa)	Knee Osteoarthritis

## Data Availability

Data are contained within the manuscript.

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
