# Peer review of "Intra-Articular Hyaluronic Acid in Osteoarthritis and Tendinopathies: Molecular and Clinical Approaches"

_biomedicines, 2023, doi:10.3390/biomedicines11041061_

Round 1

Reviewer 1 Report

I read this manuscript with title "Intra-articular hyaluronic acid in Osteoarthritis and Tendinopathies: molecular and clinical approaches" very curefully. Unfortunately I feel that this manuscript does not add somthing new in the literature.  

Author Response

Dear reviewer #1, thank you for reviewing our manuscript.

While you did not provide any suggestions for improvements we still thank you. Changes have been made in order to improve the manuscript.

Reviewer 2 Report

I suggest to add one part regarding to compare HA to other drugs used in the same diseases. Also would be good to asses cost-effectiveness of such treatment.

Author Response

Modifications have been made, including comparison with other drugs and cost-effectiveness.

Reviewer 3 Report

The review paper describes the effects of HA in OA and tendinopaties. Clinical approaches are described in more detail than molecular aspects.

Comments

1.      Line 83: The authors should indicate that the studies they describe are related to in vitro experiments as MMP-1 is not produced by chondrocytes in vivo.

2.      Lines 95-101: Articular cartilage ECM has much more complex composition. This part should be rewritten in more detail indicating types of cartilage collagens.

3.      Lines 103-119: The authors should describe all the contemporary view point on OA development in addition to “wear and tear” theory, which is now considered outdated.

4.      Section 2:  Indeed etiopathogenesis of OA should be described in more detail.

5.      Lines146-156: In this paragraph the authors should describe tendon molecular structure in more detail indicating, for example, types of collagen [PMID 24088220].

6.      Lines 161-162: This sentence contains no information. Signaling pathways and their effects on ECM function and composition should be described in detail.

7.      Lines 242; 362: Collagen type should be always indicated.

8.      Lines 253-254: The authors of reference [70] only indicate that HA can modify homeostatic activity but not reverse. This should be corrected.

9.      Lines 273-274; 295-296; 307-309: These sentences are not clear. They should be clarified.

10.  Line 309: Reference [85] is incorrect. This should be corrected.

11.  Table 1: References for each of HA formulations should be included to the Table.

12.  Line 371: The reference at the end of this sentence is required.

13.  Overall: The authors should subsections in each sections indicating the subject which they describe.

Author Response

We have applied modifications to the manuscript in order to improve it.

  1. Adjustments have been made to indicate the nature of the studies described.
  2. More information has been added.
  3. More information has been added.
  4. Etiopathogenesis of OA has been described according to previously published studies in the literature.
  5. Thank you for sharing this information, it has been added to the text in order to further describe the tendon molecular structure.
  6. Information has been added.
  7. Thank you for bringing this to our attention. We indicated the type of collagen.
  8. The title of the paper says “Hyaluronic Acid Reverses the Abnormal Synthetic Activity of Human Osteoarthritic Subchondral Bone Osteoblasts.”
  9. We tried to fix these lines in order to eliminate confusion.
  10. We removed that sentence completely as we felt it was no longer necessary and just created confusion.
  11. We added information in the text explaining that the table was created according to the research developed by Migliore et al. (DOI: 10.4137/CMAMD.S38857). All of the HA products listed were described in their study and it is now given proper citation.
  12. References were added.
  13. Thank you for your suggestions, we will try to improve the overall quality of the manuscript.

Round 2

Reviewer 3 Report

I have no more comments. Accept as is.

Author Response

Spell check and grammatical errors have been rectified as per reviewer's comments